# Psychosis proneness, loneliness, and hallucinations in nonclinical individuals

**Sarah Hope Lincoln** [1]*, **Taylor Johnson**[1], **Sarah Kim**[1], **Emma Edenbaum**[2], **Jill M. Hooley**[3]

1 Department of Psychological Sciences, Case Western Reserve University, Cleveland, OH, United States of America, 2 Department of Psychology, Oberlin College and Conservatory, Oberlin, OH, United States of America, 3 Department of Psychology, Harvard University, Cambridge, MA, United States of America

* sarahhope.lincoln@case.edu

## Abstract

Hallucinations occur along a continuum of normal functioning. Investigating the factors related to this experience in nonclinical individuals may offer important information for understanding the etiology of hallucinations in psychiatric populations. In this study we test the relationship between psychosis proneness, loneliness, and auditory hallucinations in a nonclinical sample using the White Christmas paradigm. Seventy-six undergraduate students participated in this study. We found that slightly more than half of our participants endorsed a hallucinatory experience during the White Christmas paradigm. However, we did not observe a relationship between the number of hallucinatory experiences and schizotypy, propensity to hallucinate, or loneliness. Moreover, there were no differences on these measures between individuals who reported hearing a hallucination during the White Christmas paradigm relative to those who did not. Thus, there may be other contextual factors not investigated in this study that might clarify the mechanism by which auditory hallucinations are experienced in a nonclinical population.

## Introduction

Auditory hallucinations (a sensory experience of hearing something in the absence of any actual auditory stimulus) are not limited exclusively to individuals with psychotic disorders [1]. Rather, hallucinations occur along a continuum of normal functioning and are reported in nonclinical individuals [2, 3]. Approximately 9.6% of the general population endorses having a hallucination their lifetime [4]. What is less well understood is the mechanism that may explain why some healthy individuals hear hallucinations and others do not.

Previous research suggests that individual differences across a variety of factors explain some of the variance in hallucinatory experiences. Emotional states, for example high levels of negative affect, are associated with a propensity to hear hallucinations [5]. Cognitive factors, such as positive beliefs about unusual experiences [6, 7], are also associated with a propensity to hallucinate. Other work suggests that fantasy-proneness [8–10] is associated with a hallucination predisposition. Individual differences in schizotypy may also explain variance in hallucinatory experiences. Schizotypy is a personality dimension that indicates a propensity to develop schizophrenia, and may reflect a psychosis phenotype [11]. Schizotypy has three dimensional factors which correspond with the positive, negative, and disorganized symptoms

**Competing interests:** The authors have declared
that no competing interests exist.

in schizophrenia [12]. Schizotypal characteristics include magical thinking, paranoia, unusual
perceptual experiences, odd beliefs and behavior, disorganized speech, and social anhedonia.
Schizophrenia spectrum disorders and schizotypy share commonalities across neuropsycho-
logical, social and environmental, and biological factors [13].

Though individual differences may predispose some people to hallucinations other research
suggests that situational context is also important. In particular, a diathesis-stress model sug-
gests that individuals with a predisposition for hallucinations are more likely to experience hal-
lucinations in the context of stress. For example, in a signal detection task in which individuals
were told to listen for speech embedded in white noise (even though no speech was actually
embedded in the white noise), people with high levels of trait anxiety reported more false
alarms (*i.e.*, hallucinations) in stressful versus non-stressful conditions [14]. In another study,
Crowe and colleagues [15] found that individuals with high levels of recent caffeine intake
were more likely to report an auditory hallucination in the context of stress relative to no-
stress. Although preliminary, research such as this highlights the importance of context in the
experience of hallucinations.

One way of exploring the types of contexts in which hallucinations may be reported by the
general population is through the White Christmas paradigm. This is a type of signal detection
task where individuals are asked to indicate if a signal is present. The task involves telling par-
ticipants that they will be hearing white noise and that fragments of the Bing Crosby song
*White Christmas* may be embedded in the white noise. In actuality, the song is never played
[8]. In this type of task any false positive is considered a hallucination because it is the detec-
tion of a signal in the absence of an actual signal [14]. The White Christmas paradigm has
been repeatedly used to test the propensity to hallucinate in different nonclinical samples. For
example, Crowe et al. [15] used this paradigm when examining the role of caffeine and stress
on auditory hallucinations in healthy adults, finding that only in the presence of both high caf-
feine intake and high stress did healthy adults report auditory hallucinations.

In the current study, using a modification of the White Christmas paradigm, we tested the
experience of auditory hallucinations in the context of loneliness. Loneliness is a specific
stressor that has been linked to psychotic-like experiences Previous work suggests that aspects
of social isolation, like living alone or feeling lonely, are associated with emergence of psy-
chotic-like symptoms [16]. The association between loneliness and hallucinations has been
demonstrated across several different disorders including borderline personality disorder [17],
Alzheimer's disease [18], and psychosis [19]. Additionally, in the presence of loneliness, psy-
chotic-like symptoms increased for individuals with high levels of negative and disorganized
schizotypal traits [20]. This finding supports the argument that the context of loneliness might
increase the likelihood of psychotic-like experiences in already vulnerable individuals.

Based on past research using the White Christmas paradigm, we expected that a portion of
participants would report hallucinations (*i.e.*, false alarms) during this task. We hypothesized
that higher scores on measures of psychosis-proneness such as schizotypy and propensity to
hallucinate would be correlated with a greater number of reported hallucinations. Addition-
ally, we expected to find a significant relationship between psychosis-proneness and loneliness.
Finally, we hypothesized that loneliness would mediate the relationship between the number
of hallucinations and psychosis-proneness.

## Methods

### Participants

Participants were recruited from two universities ($N$ = 76, 42 females) with a mean age of
19.68 years ($SD$ = 1.38). Approximately 56.6% of the sample identified as white, 6.6% as

African American, 26.3% as Asian, 10.5% as having another racial background, and 13.2% identifying as having a Latino/Hispanic identity. Participants were recruited through the psychology departments' research participation systems. Individuals received course credit for participation in the study. Participants were told that this study was about social and auditory perceptions; they were not aware of the study's intent to elicit hallucinatory experiences, thus preventing response bias in our sample.

## Measures

Loneliness was assessed by the R-UCLA [21]. The R-UCLA measure is a 20-item scale designed to measure self-reported loneliness and consists of a series of first-person statements regarding companionship and feelings of belonging. Higher scores indicate greater loneliness.

The presence of schizotypy was assessed by the 38-item Multidimensional Schizotypy Scale–Brief (MSS-B) [22]. The MSS-B assesses positive, negative, and disorganized schizotypy traits, via 13 positive, 13 negative, and 12 disorganized items. Sample items from the positive, negative, and disorganized subscales are as follows, respectively: "I have sometimes felt that strangers were reading my mind," "I have always preferred to be disconnected from the world," and "Most of the time I find it is very difficult to get my thoughts in order." Participants respond to first-person statements on a true-false scale.

The 13-item Revised Hallucination Scale (RHS) [6] was used to measure proneness to a range of hallucinatory experiences including vivid thoughts, intrusive thoughts, vivid daydreams, auditory hallucinations, and visual hallucinations. Participants respond to first-person statements paired with a true-false scale. The RHS items were adapted from the original Launay-Slade Hallucination Scale [23].

Preliminary pilot testing suggested that the song *White Christmas* was not highly familiar to our sample population of college undergraduates. We therefore adapted the White Christmas paradigm [8] for use in this study and used the song *Somewhere Over the Rainbow* sung by Judy Garland. This song was better known to students in our pilot testing. Low white noise from the soundtrack *White Noise*: *Loopable White Noise Sounds for Sleep and Relaxation* was played for three minutes.

## Procedure

This study was approved by the Case Western Reserve University Institutional Review Board. On arrival at the lab, participants first provided written consent. They then completed the modified White Christmas paradigm. First, participants listened to the song *Somewhere Over the Rainbow*. Then participants were told that they would listen to a white noise track via headphones for approximately three minutes. Before listening to the white noise, participants were given the following instructions based on the prompt from the Merckelbach and van de Ven [8] study:

> "The *Somewhere Over the Rainbow* song you just heard might be embedded in the white noise below the auditory threshold. If you think or believe that you hear the song clearly, please press the button in front of you. Of course, you may press the button several times if you think that you heard several fragments of the song."

Participants were instructed to press the "y" key on the keyboard if at any time they believed they heard fragments of *Somewhere Over the Rainbow* embedded within the white noise. After completing the White Christmas task, the participants completed all self-report questionnaires.

**Data analysis.** For this study we planned to first conduct correlation analyses assessing the relationships between number of hallucinations and both schizotypy and propensity to hallucinate. If a significant relationship is found between either the number of hallucinations and schizotypy or the number of hallucinations and propensity to hallucinate, we will conduct mediation analyses, testing loneliness as the mediator between these variables.

## Results

Even though participants were not selected for psychopathology a majority of our sample, forty-five participants (59.2%), reported a false alarm, or hallucination. The number of reported hallucinations ranged from 1 to 16 with a mean of 2.59 hallucinations per participant. Means and standard deviations for all questionnaires are reported in Table 1.

We hypothesized that the number of hallucinations reported by each individual would be associated with their self-reports of schizotypy and propensity to hallucinate, and that loneliness would mediate these relationships. Because skew and kurtosis contributed to lack of normality in our data, we performed a base 10 logarithm transformation of our variables. In total we ran seven correlations. Using a Bonferroni correction to adjust for multiple tests this means that a $p$ value of .007 is required for significance. Our analyses revealed no association between the number of hallucinations and self-reports of schizotypy ($r = .04$, $p = .731$), propensity to hallucinate ($r = .11$, $p = .403$), or loneliness ($r = 0.02$, $p = .878$). Because there was not a statistically significant relationship between our variables of interest, a mediation analyses was no longer appropriate for our data.

To further confirm the lack of significant associations, in a post-hoc analyses we divided our participants into two groups—individuals who reported no hallucinations versus individuals who reported hallucinations (meaning that they heard the song in the white noise). We hypothesized that individuals who reported hallucinations would have higher schizotypy scores, a greater propensity to hallucinate, and higher reports of loneliness. T-tests revealed no difference between these two groups in regard to schizotypy ($M_1 = 3.43$, $M_2 = 3.01$, $t = 1.24$, $p = .219$), propensity to hallucinate ($M_1 = 2.80$, $M_2 = 2.32$, $t = .55$, $p = .583$) or loneliness ($M_1 = 35.26$, $M_2 = 36.41$, $t = .60$, $p = 553$). We also compared participants in the upper and lower quartiles of hallucination reports. As before, there was no difference between these groups of individuals on schizotypy ($t = -.40$, $p = .69$), propensity to hallucinate ($t = .57$, $r = .57$), or loneliness ($t = -1.12$, $p = .27$). Our primary hypothesis, that the frequency of hallucinations would be associated with psychosis-proneness and loneliness was not supported.

Finally, we examined the hypothesis that individuals who experienced higher rates of loneliness would also have higher rates of schizotypy. We found a significant positive correlation between loneliness and schizotypy, such that individuals who reported higher rates of loneliness also endorsed more schizotypal traits ($r = .40$, $p < .001$). Specifically, loneliness was

**Table 1. Means and standard deviations of the White Christmas Paradigm, Multidimensional Schizotypy Scale (MSS), Revised Hallucination Scale, and the UCLA Loneliness Scale.**

| Questionnaires | Mean (Std Dev) $N = 76$ |
|---|---|
| White Christmas Paradigm (# of button-presses) | 2.59 (3.46) |
| MSS Total | 3.18 (3.38) |
| MSS Positive | 1.10 (1.65) |
| MSS Negative | .93 (1.38) |
| MSS Disorganized | 1.15 (2.03) |
| Revised Hallucination Scale | 2.51 (1.94) |
| UCLA Loneliness Scale | 35.93 (8.13) |

significantly associated with scores on the negative subscale ($r = .51$, $p < .001$) and the disorganized subscale ($r = .31$, $p = .007$), but not the positive subscale ($r = .002$, $p = .990$) on the Multidimensional Schizotypy Scale-Brief. These correlations remained even when social anhedonia items from the Multidimensional Schizotypy Scale that might overlap with the items on the UCLA Loneliness scale were removed.

## Discussion

We investigated the extent to which loneliness and schizotypy were associated with hallucinations in a healthy undergraduate sample. Using a modification of the White Christmas paradigm, we found that over half of our participants (59.2%) reported at least one hallucination experience during this task. This figure is slightly higher than findings from previous studies where approximately 32% [8] and 35% [9] of undergraduates reported hallucinations.

Although we hypothesized a relationship between the number of hallucinations and both schizotypy scores and a propensity to hallucinate we found no such association. Our hypothesis that loneliness would mediate these relationships was therefore not supported. In follow-up analyses we also found no significant differences in schizotypy scores, propensity to hallucinate, or loneliness between individuals who reported hallucinations relative to those who did not report hallucinations.

It should be noted that the percentage of people reporting hallucinations during our task is large enough to suggest that there is some phenomenon occurring and these hallucinatory experiences are more than noise in the data. As such, there are several possible reasons that we did not find this expected relationship. It is possible that the demands of the task override the actual experience of hallucinations, with people feeling a need to "perform" well on the task and report a fake hallucination. Perhaps if we had looked at the degree of confidence with which people reported the hallucinations, we may have obtained a more accurate count of those who genuinely believed they experienced a hallucination. Additionally, we may not have seen the relationship between hallucinations and psychosis-proneness because our sample had a limited range of schizotypy scores. The Multidimensional Schizotypy Scale-Brief has a total possible score of 38, and our sample fell within a range of zero to 15. Perhaps the relationship we were attempting to detect occurs only in individuals with higher schizotypy scores and our truncated sample did not capture these individuals. Finally, it may be that the relationship between psychosis-proneness and hallucination occurs only in the presence of an acute stressor, as found in the Crowe et al. [15] study. Loneliness may not have functioned as a significant acute stressor in our model.

Of note, we did find a significant positive relationship between schizotypy and loneliness scores. Undergraduate students who endorsed more schizotypal items also reported more loneliness in their day-to-day lives. This is consistent with previous research indicating that loneliness is related to psychotic-like experiences and is a risk factor for the development of psychotic disorders [19, 20]. Additionally, this finding might have important implications for interventions on loneliness with individuals who might be at risk for psychosis [24].

In light of the lack of expected relationship between number of hallucinations and reports of psychosis-proneness and loneliness, we remain curious about what factors and contexts might be associated with hallucinations in nonclinical individuals. The White Christmas paradigm predictably elicits hallucinations in a sizable portion of individuals [8, 9, 15] making this a promising approach for understanding hallucinations in a nonclinical sample. In this study we investigated whether loneliness might explain a relationship between psychosis-proneness and hallucinations. We explored the data in a number of different ways, increasing our confidence that this relationship does not exist in our sample. Future research should investigate

individuals with higher rates of schizotypy traits and potentially individuals at clinical high risk for psychotic disorders; these people may be more vulnerable for the experience of hallucinations. We should note that we modified the original paradigm in two ways. First, we did not use the song *White Christmas*. Second, white noise can vary a great deal in its frequencies and intensities; we do not know what white noise was used in the original paradigms and it is possible their white noise differs from the white noise we used. However, we do not think these two modifications have an effect on our data, as we see approximately the same percentage of individuals reporting hallucinations in the white noise as with previous studies. We may also be limited by examining these phenomenon in a college student sample. An additional limitation is that the correlational nature of this study did not allow us to look at the mechanism by which the phenomenon may be occurring. Our analyses do not directly assess the mechanism. Additionally, the hypothesized relationships may only occur in situations in which there is an acute stressor and future research could test for the relationship between hallucinations and psychosis-proneness in the presence of an acute social stressors such as rejection or exclusion. Ultimately, the applicability of this paradigm may be for a different combination of risk factors that have yet to be tested.

## Author Contributions

**Conceptualization:** Sarah Hope Lincoln, Jill M. Hooley.

**Data curation:** Sarah Hope Lincoln, Taylor Johnson, Sarah Kim.

**Formal analysis:** Sarah Hope Lincoln, Taylor Johnson, Sarah Kim.

**Investigation:** Sarah Hope Lincoln, Jill M. Hooley.

**Methodology:** Sarah Hope Lincoln, Jill M. Hooley.

**Project administration:** Sarah Hope Lincoln, Taylor Johnson.

**Supervision:** Jill M. Hooley.

**Writing – original draft:** Sarah Hope Lincoln, Taylor Johnson, Sarah Kim, Emma Edenbaum, Jill M. Hooley.

**Writing – review & editing:** Sarah Hope Lincoln, Taylor Johnson, Sarah Kim, Emma Edenbaum, Jill M. Hooley.

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
