## [Decision Letter · Decision Letter 0]

4 Mar 2021

PONE-D-20-39171

Psychosis Proneness, Loneliness, and Hallucinations in Nonclinical Individuals

PLOS ONE

Dear Dr. Lincoln,

Thank you for submitting your manuscript to PLOS ONE. After careful consideration, we feel that it has merit but does not fully meet PLOS ONE’s publication criteria as it currently stands. Therefore, we invite you to submit a revised version of the manuscript that addresses the points raised during the review process.

I was fortunate to receive reviews from two experts in this area. I thank them for their attention to this manuscript. The reviewers identify that there are important contributions of the work. However, they also note some areas of the manuscript that could be clearer in presentation. Some of these may lead to additional presentation of analyses for clarity. These may best be accomplished through inclusion of bivariate correlations among study variables. Reviewers note that some questions about the inclusion/testing of mediation (rather than moderation); I think that it is also critical to refrain from language of mediation in light of the cross-sectional design. Reviewer 1 also noted areas of the conceptual foundation of the work that could be elaborated on in the introduction.

Beyond the issues noted by the Reviewers, I also wanted to highlight a need to describe magnitude of effects reported by previous studies using the White Christmas paradigm. The corresponding need is to qualify findings in light of the study’s ability to recover those effects. I am not asking for a post-hoc power analysis based on this study’s finding, but to identify the minimum effect that could have been found given the study design. Conversely, as the conclusions are written in a way that is “supporting the null hypothesis,” recent methods have been developed to show that associations are smaller than what would be a meaningful effect, such as using equivalence testing (e.g., Lakens, Scheek, & Isager, 2018; https://doi.org/10.1177/2515245918770963).

We look forward to receiving your revised manuscript.

Kind regards,

Thomas M. Olino

Academic Editor

PLOS ONE

Journal Requirements:

2. Please note that peer review at PLOS ONE is not double-blinded (https://journals.plos.org/plosone/s/editorial-and-peer-review-process).

For this reason, authors should include in the revised manuscript all the information that may have been removed for blind review, including names of the universities from which participants were recruited.

Reviewers' comments:

Reviewer's Responses to Questions

**Comments to the Author**

1. Is the manuscript technically sound, and do the data support the conclusions?

Reviewer #1: Partly

Reviewer #2: Yes

2. Has the statistical analysis been performed appropriately and rigorously? 

Reviewer #1: Yes

Reviewer #2: Yes

3. Have the authors made all data underlying the findings in their manuscript fully available?

Reviewer #1: Yes

Reviewer #2: No

4. Is the manuscript presented in an intelligible fashion and written in standard English?

Reviewer #1: Yes

Reviewer #2: Yes

5. Review Comments to the Author

Reviewer #1: Manuscript Number: PONE-D-20-39171

Title: Psychosis Proneness, Loneliness, and Hallucinations in Nonclinical Individuals

Introduction

This is an interesting manuscript on the relationships between psychosis proneness, loneliness, and auditory hallucinations in a nonclinical sample using an adaptation of the well-validated paradigm, White Christmas. The authors provide sound justification for investigating this question. In terms of findings, while the authors found that over half the sample reported at least one hallucination experience in response to the task, no correlations were observed between the number of hallucinations, schizotypy, and propensity to hallucinate. While the study is overall clearly written, further clarification regarding the statistical analyses are needed, particularly whether mediation was used. For example, the authors describe the aim of investigating whether loneliness mediated relationships, but it is unclear what types of mediation analyses were employed, if any. Overall, this manuscript is well-constructed and offers contributions to the field in this area. Please see major and minor issues noted below.

Major Issues

It would be useful to have a more in depth description of nonclinical psychosis and schizotypy earlier on for readers that may not be as familiar with experiences endorsed by this group.

The authors might consider discussing, in the limitations section perhaps, that the correlational approach to the study does not necessarily get at mechanism, but instead hints towards processes involved. The authors discuss unknown mechanistic understanding of these processes in the introduction and clarification that the analyses employed are not directly examining mechanism can more accurately help with interpreting correlational findings.

How did the authors consider and account for site differences between the two universities? Did you conduct any analyses to support combining the samples?

The authors might consider adding in statistical analyses to support the written phrase, “This song was better known to students in our pilot testing” described in the methods section.

The authors discuss statistical analyses generally, but it appears there is not a data analysis section describing which tests were employed. For example, the authors discuss examining mediation in the sample, but it is unclear how this was examined statistically (“we hypothesized that loneliness would mediate the relationship between the number of hallucinations and psychosis-proneness” pg. 5). Did the authors use mediation statistics? Along these lines, what type of log transformation did the authors use?

The inclusion of the limitation of utilizing a purely undergraduate sample should be discussed briefly.

Minor Issues

There are minor spelling and grammar mistakes throughout (e.g. pg. 4, I think you mean current study, also on page 4, a period I missing after the second sentence in the last paragraph, a comma missing in the last paragraph on pg. 3). Furthermore, in some places, the authors write psychotic like and psychotic-like; consistency in this can enhance readability.

Reviewer #2: The authors examined relations between loneliness, hallucinations, and schizotpy in a nonclinical sample of undergraduate students. The article is well-written and clear with an appropriate discussion of the null findings in the context of the study’s limitations. There are only few minor suggestions/questions for improvement of the manuscript:

1) Why did the authors hypothesize a mediation rather than a moderation model for the effect of loneliness on psychosis proneness and hallucinations? A justification for this hypothesis is needed.

2) The social desirability measure in the method section comes out of nowhere and there is no explanation for why it is included. It is also never referenced after the method section. The authors should remove this from the manuscript or provide a rationale for its inclusion and relevant findings.

3) The authors may want to discuss adapting the paradigm as a limitation and speculate on whether or not this could have influenced their findings.

6. PLOS authors have the option to publish the peer review history of their article (what does this mean?). If published, this will include your full peer review and any attached files.

Reviewer #1: No

Reviewer #2: No

---

## [Author Response · Author response to Decision Letter 0]

29 Apr 2021

April 26, 2021

RE: PONE-D-20-39171

Dear Dr. Olino,

Thank you for your careful review and constructive comments provided on our manuscript “Psychosis Proneness, Loneliness, and Hallucinations in Nonclinical Individuals.” To further strengthen our submission, we have addressed each of the additional concerns raised, and we believe the review process has improved the overall quality of our submission. Additionally, we have looked at equivalence testing for this paper and have documented those results in response to your feedback below. We welcome any further discussion or questions of that method; it was new to us and interesting to learn about. Thank you for your time and efforts in considering our revised manuscript.

Comment from Editor

Beyond the issues noted by the Reviewers, I also wanted to highlight a need to describe magnitude of effects reported by previous studies using the White Christmas paradigm. The corresponding need is to qualify findings in light of the study’s ability to recover those effects. I am not asking for a post-hoc power analysis based on this study’s finding, but to identify the minimum effect that could have been found given the study design. Conversely, as the conclusions are written in a way that is “supporting the null hypothesis,” recent methods have been developed to show that associations are smaller than what would be a meaningful effect, such as using equivalence testing (e.g., Lakens, Scheek, & Isager, 2018; https://doi.org/10.1177/2515245918770963).

Thank you for this comment. We have gone back and calculated the effect sizes from previous papers, specifically van de Ven & Merckelbach (2003) and Merckelbach & van de Ven (2001). In Merckelbach & van de Ven (2001), 32 % of the people reported a hallucination, and they found a significant difference between those who heard the hallucinations and those who did not on the Launay-Slade Hallucination Scale with a moderate effect size (d = 0.62). In van de Ven & Merckelbach (2003) they did not find a difference between groups on either the Schizotypal Personality Scale (d = .25) or the Launay-Slade Hallucination Scale (d = .41). In looking at our data, with our sample size, we have 76% power to detect a moderate effect size, but only 20% power to detect a smaller effect. 

In the article you reference (Lakens et al., 2018) they talk about picking the Smallest Effect Size of Interest (SESOI); they suggest starting with the “largest effect size that, when observed in the original study, would not have been statistically significant.” In this instance, the effect size d = .41 is the largest non-significant effect size. In using the Two One Sided T-tests (TOST) for correlations we find that our confidence interval does fall within the upper and lower bounds and thus is not significantly different from zero (Figure 1). At the same time, given that we would want the White Christmas paradigm to potentially be an indicator of a vulnerability to psychosis, we would likely want a larger effect in order for it to be clinically meaningful. Because we had previously not been familiar with equivalence testing, we have not included these results in our manuscript, but in deference to your comment we have stated in the discussion Though we failed to find evidence of an association that does not mean that no association exists. 

Figure 1. Number of hallucinations correlated with schizotypy score. Our confidence interval falls within the equivalence bounds.

Reviewer #1

1. It would be useful to have a more in depth description of nonclinical psychosis and schizotypy earlier on for readers that may not be as familiar with experiences endorsed by this group.

Thank you for this suggestion. We have now included a paragraph about schizotypy in the introduction to hopefully make this construct clearer to readers.

Schizotypy is a personality dimension that indicates a propensity to develop schizophrenia, and may reflect a psychosis phenotype (Van Os, Linscott, Myin-Germeys, Delespaul, & Krabbendam, 2009). Schizotypy has three dimensional factors which correspond with the positive, negative, and disorganized symptoms in schizophrenia (Nelson, Seal, Pantelis, & Phillips, 2013). Schizotypal characteristics include magical thinking, paranoia, unusual perceptual experiences, odd beliefs and behavior, disorganized speech, and social anhedonia. Schizophrenia spectrum disorders and schizotypy share commonalities across neuropsychological, social and environmental, and biological factors (Neal et al., 2013). 

2. The authors might consider discussing, in the limitations section perhaps, that the correlational approach to the study does not necessarily get at mechanism, but instead hints towards processes involved. The authors discuss unknown mechanistic understanding of these processes in the introduction and clarification that the analyses employed are not directly examining mechanism can more accurately help with interpreting correlational findings.

Thank you for this suggestion; you’re right, we do not look at mechanisms. We have added in the discussion the following sentences: 

An additional limitation is that the correlational nature of this study did not allow us to look at the mechanism by which the phenomenon may be occurring. Our analyses do not directly assess the mechanism.

3. How did the authors consider and account for site differences between the two universities? Did you conduct any analyses to support combining the samples?

Thank you for this suggestion. We have run tests comparing the two groups. There is no difference for age (p = .061), schizotypy total scores (p = .447), loneliness (p = .844), and propensity to hallucinate (p = .168). Nor was there a difference on the number of people who heard hallucinations through the white noise (p = .644). 

4. The authors might consider adding in statistical analyses to support the written phrase, “This song was better known to students in our pilot testing” described in the methods section.

Thank you for this suggestion. We piloted the study with the undergraduate students in our lab at the time (n ~ 8), but unfortunately we did not collect data on these individuals.

5. The authors discuss statistical analyses generally, but it appears there is not a data analysis section describing which tests were employed. For example, the authors discuss examining mediation in the sample, but it is unclear how this was examined statistically (“we hypothesized that loneliness would mediate the relationship between the number of hallucinations and psychosis-proneness” pg. 5). Did the authors use mediation statistics? Along these lines, what type of log transformation did the authors use?

Thank you for this comment. We have now added a data analysis section that reads: 

For this study we will first conduct correlation analyses assessing the relationships between number of hallucinations and both schizotypy and propensity to hallucinate. If a significant relationship is found between either the number of hallucinations and schizotypy or the number of hallucinations and propensity to hallucinate, we will conduct mediation analyses, testing loneliness as the mediator between these variables. 

We have further clarified that mediation analyses were not conducted, because we did not observe a statistically significant relationship between number of hallucinations and either schizotypy or propensity to hallucinate. There was no relationship to mediate.

Additionally, we used a log10 for our log transformation, we have included this information in the results section, but not in the data analysis plan, as we did not a priori plan to do log transformations.

6. The inclusion of the limitation of utilizing a purely undergraduate sample should be discussed briefly.

Thank you for this suggestion. We have added the following sentence to the discussion section. 

We may also be limited by examining these phenomenon in a college student sample.

7. There are minor spelling and grammar mistakes throughout (e.g. pg. 4, I think you mean current study, also on page 4, a period I missing after the second sentence in the last paragraph, a comma missing in the last paragraph on pg. 3). Furthermore, in some places, the authors write psychotic like and psychotic-like; consistency in this can enhance readability.

Thank you for these notes. We have gone through a proofread the paper to correct for any grammatical errors and we have changed “psychotic like” to “psychotic-like” throughout the paper.

Reviewer #2

8. Why did the authors hypothesize a mediation rather than a moderation model for the effect of loneliness on psychosis proneness and hallucinations? A justification for this hypothesis is needed.

Thank you for this comment. We hypothesized a mediation because we thought that loneliness would explain the relationship between the two variables rather than have an effect on its strength or direction. This idea is based off of the social deafferentation hypothesis which suggests that lack of social input may lead to psychotic symptoms like hallucinations. We thought that loneliness, as a proxy for lack of social input, would explain why there was a relationship between hearing music through the white noise and the schizotypy scores. We did not end up running mediation analyses because we did not see a significant relationship between hallucinations and either schizotypy scores or the revised hallucination scale; thus there was no relationship to mediate.

9. The social desirability measure in the method section comes out of nowhere and there is no explanation for why it is included. It is also never referenced after the method section. The authors should remove this from the manuscript or provide a rationale for its inclusion and relevant findings.

Thank you for pointing this out. We have removed reference to the social desirability scale as we did not use this in our reported analyses.

10. The authors may want to discuss adapting the paradigm as a limitation and speculate on whether or not this could have influenced their findings.

Thank you for this suggestion. We have added the following sentences to our discussion section.

We should note that we modified the original paradigm in two ways. First, we did not use the song White Christmas. Second, white noise can vary a great deal in its frequencies and intensities; we do not know what white noise was used in the original paradigms and it is possible their white noise differs from the white noise we used. However, we do not think these two modifications have an effect on our data, as we see approximately the same percentage of individuals reporting hallucinations in the white noise as with previous studies.

Again, thank you for your careful review and consideration of our manuscript. We truly believe the Reviewers’ suggestions have contributed to an improved product.

Sincerely,

Sarah Hope Lincoln, Ph.D.

---

## [Editor Report · Decision Letter 1]

3 May 2021

Psychosis Proneness, Loneliness, and Hallucinations in Nonclinical Individuals

PONE-D-20-39171R1

Dear Dr. Lincoln,

I reviewed your responses and revision and saw that you were fully responsive to the previous comments. I do not see a need to have the manuscript sent out for an additional review. We’re pleased to inform you that your manuscript has been judged scientifically suitable for publication and will be formally accepted for publication once it meets all outstanding technical requirements. 

Kind regards,

Thomas M. Olino

Academic Editor

PLOS ONE
---

## [Editor Report · Acceptance letter]

20 May 2021

PONE-D-20-39171R1 

Psychosis Proneness, Loneliness, and Hallucinations in Nonclinical Individuals 

Dear Dr. Lincoln:

I'm pleased to inform you that your manuscript has been deemed suitable for publication in PLOS ONE. Congratulations! Your manuscript is now with our production department. 

Kind regards, 

on behalf of

Dr. Thomas M. Olino 

Academic Editor

PLOS ONE